# Crystal structure of Zen4 in the apo state reveals a missing conformation of kinesin

Ruifang Guan[1,2,*], Lei Zhang[3,*], Qian Peter Su[4], Keith J. Mickolajczyk[5], Geng-Yuan Chen[5], William O. Hancock[5], Yujie Sun[4], Yongfang Zhao[3] & Zhucheng Chen[1,2]

Kinesins hydrolyse ATP to transport intracellular cargoes along microtubules. Kinesin neck linker (NL) functions as the central mechano-chemical coupling element by changing its conformation through the ATPase cycle. Here we report the crystal structure of kinesin-6 Zen4 in a nucleotide-free, apo state, with the NL initial segment (NIS) adopting a backward-docked conformation and the preceding α6 helix partially melted. Single-molecule fluorescence resonance energy transfer (smFRET) analyses indicate the NIS of kinesin-1 undergoes similar conformational changes under tension in the two-head bound (2HB) state, whereas it is largely disordered without tension. The backward-docked structure of NIS is essential for motility of the motor. Our findings reveal a key missing conformation of kinesins, which provides the structural basis of the stable 2HB state and offers a tension-based rationale for an optimal NL length to ensure processivity of the motor.

[1] MOE Key Laboratory of Protein Science, Tsinghua University, Beijing 100084, China. [2] School of Life Science, Tsinghua University, Beijing 100084, China. [3] National Laboratory of Biomacromolecules, Institute of Biophysics, Beijing 100101, China. [4] State Key Laboratory of Membrane Biology, Biodynamic Optical Imaging Center (BIOPIC), School of Life Sciences, Peking University, Beijing 100871, China. [5] Department of Biomedical Engineering, Pennsylvania State University, University Park, Pennsylvania 16802, USA. * These authors contributed equally to this work. Correspondence and requests for materials should be addressed to Z.C. (email: zhucheng_chen@tsinghua.edu.cn).

Kinesins are a large family of molecular motor proteins that utilize the energy from ATP hydrolysis to move along microtubules and transport various cellular cargoes[1]. Conventional kinesin (kinesin-1) is the founding member of the kinesin family of proteins and has been extensively studied[2]. Kinesin-1 is a dimeric motor that exhibits high processivity, taking over 100 steps along a microtubule before disassociation[3,4].

A key intermediate in the stepping cycle of kinesin is the two-head bound (2HB) state, in which the leading head is in the apo state and the trailing head is in the ATP/ADP-Pi-bound state[5,6]. Kinesin-1 spends most of its time in the 2HB state at the physiological concentration of ATP[7,8], during which ATP is hydrolysed and inorganic phosphate is released from the trailing head[5]. The two bound heads span a distance of $\sim 8$ nm and are connected together through the NL and the neck coiled-coil helix[9]. In this configuration, the NL is stretched and intramolecular tension is generated between the two bound heads. The intramolecular tension is widely believed to be the key for the head–head communication to ensure the processivity of the motor[8,10–13].

The intramolecular tension correlates with the structure of NL. The NL favours a forward-docked conformation in ATP-bound state, whereas it preferentially adopts a disordered structure in the ADP-bound state[14]. Numerous crystal and cryoEM structures of the motor domain in complex with ATP and ADP have been obtained in support of this idea[15–19]. However, very few structures of kinesin in the apo state have been reported[20–22], as the motor domain by itself is unstable in the absence of nucleotide[23].

Although it cannot be easily measured, the magnitude of intramolecular tension can be estimated by the worm-like chain (WLC) model, which has been shown to faithfully recapitulate the force-extension curves of unfolded polypeptide and DNA[24,25]. From the WLC model, estimates for inter-head tension based on the current model range from 12–15 pN up to $\sim 28$ pN (refs 26,27). Considering the stable binding of kinesin heads in the 2HB state, these magnitudes are large relative to the $\sim 7$ pN unbinding force or stall force, and seem improbable[27–30]. One reason for this discrepancy is that remains unclear what conformation the NL and the preceding $\alpha 6$ helix adopt under tension in the apo state[27].

Here we report the crystal structure of the mitotic kinesin Zen4 in the apo state, in which the end of the $\alpha 6$ helix is unwound and the NIS adopts a backward-docked conformation. This structure motivates a reexamination of the NIS of kinesin-1 and a re-evaluation of the 2HB kinesin structure.

## Results

**Structure of Zen4 in the apo state.** We determined the structure of a mitotic kinesin Zen4 from *C. elegans*. Zen4, MKLP1 (human homologue) and Pavarotti (*Drosophila* homologue) are kinesin-6 family proteins. They walk to and accumulate at the plus-end of interdigitated microtubules, playing a key role in formation of antiparallel central spindles during cytokinesis in animal cells[31]. The kinesin-6 family proteins have two characteristic features: a long NL and a large insertion in the motor domain (Supplementary Fig. 1). We crystallized the motor domain of Zen4 (1–441). The final structure was refined to 2.6 Å, with $R_{work}/R_{free} = 21\%/24\%$ (Table 1). The first 24 residues at the N terminus and the last 15 residues at the C terminus are disordered in the crystals.

Zen4 adopts a typical kinesin fold. The insertion sequence emerges from the C-terminal end of $\alpha 2b$, interacts with the central $\beta$-sheet ($\beta 4$–$\beta 7$), $\alpha 3$, $\alpha 2b$, $\alpha 1$, and finally connects back to the $\beta 4$-strand (Fig. 1). Thus, the large insertion sequence glues

**Table 1 | Data collection and refinement statistics (molecular replacement).**

| | |
|---|---|
| *Data collection* | |
| Space group | P6 |
| Cell dimensions | |
| *a, b, c* (Å) | 244.152, 244.152, 42.217 |
| $\alpha, \beta, \gamma$ (°) | 90, 90, 120 |
| Resolution (Å) | 50.00–2.61 (2.70–2.61) |
| $I/\sigma I$ | 13.3 (1.05) |
| Completeness (%) | 99.6 (99.9) |
| Redundancy | 5.2 |
| CC1/2 | 0.50 |
| | |
| *Refinement* | |
| Resolution (Å) | 31.84–2.61 (2.703–2.61) |
| No. of reflections | 44,541 |
| $R_{work}/R_{free}$ | 0.2087/0.2420 |
| No. of atoms | |
| Protein | 5,812 |
| Ligand/ion | 42 |
| Water | 148 |
| *B*-factors | |
| Protein | 78 |
| Ligand/ion | 119 |
| Water | 64 |
| R.m.s. deviations | |
| Bond lengths (Å) | 0.003 |
| Bond angles (°) | 0.652 |

*Values in parentheses are for highest-resolution shell.

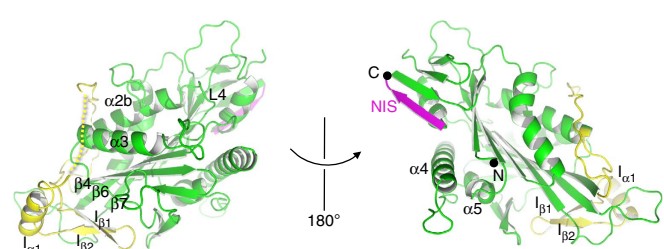

**Figure 1 | Structure of Zen4 in the nucleotide-free state.** Two different views of the overall structure of kinesin Zen4. The unique insertion sequence, yellow; the NIS, magenta. The N- and C-termini of Zen4 are labelled with black dots.

multiple structural elements together, and may function in stabilizing the structure of Zen4. The insertion sequence binds to the central $\beta$-sheet at the side distal to the microtubule-binding interface ($\alpha 4$–$\alpha 5$), suggesting the insertion motif does not interfere the interactions between the motor domain and microtubules, consistent with an earlier study[32].

Interestingly, Zen4 is in the nucleotide-free, apo state. The structure reveals an ion, rather than a nucleotide, is bound by the P-loop (L4 loop; Supplementary Fig. 2a). Comparison with the previous structure indicates that the structure of Zen4 has three features of a kinesin in the apo state[22]. First, the ATPase catalytic site of Zen4 is in an open conformation, with the Switch I region (L9) moving away from the P-loop (Supplementary Fig. 2b). Second, the N terminus of Switch II helix ($\alpha 4$) extends two helical turns and moves outward, consistent with enhanced microtubule binding in the apo state. Third, the nucleotide-binding pocket of Zen4 is occluded. In the ADP-bound state of kinesin-1, the adenine base of ADP is tightly sandwiched between Pro17 of L1 and His93 of $\alpha 2a$ through van der Waals

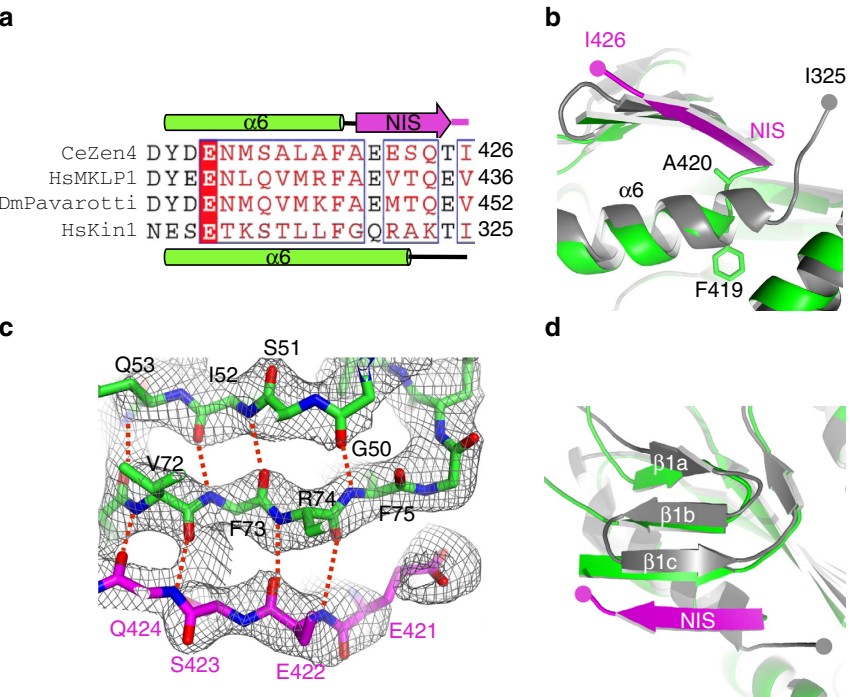

**Figure 2 | Unwound α6 and backward-docked NIS of Zen4 in the apo state. (a)** Sequence alignments around the NIS region of three kinesin-6 family proteins and human kinesin-1 (HsKin1). The secondary structure assignments on the top and at the bottom are based on the structures of Zen4 and kinesin-1 (PDB code 1BG2), respectively. **(b)** Superimposition of the structure of Zen4 with that of kinesin-1 in the ADP-bound state, showing region around the α6 helix. The alignment was done as in Fig. 1c. The grey and magenta dots indicate the last ordered residue at the C termini of kinein-1 and Zen4, respectively. **(c)** Superimposition of the structure of Zen4 around the NIS with the 2Fo-Fc map at contour level of σ = 1. **(d)** Superimposition of the structure of Zen4 with that of kinesin-1 around the N-terminal appending β-sheet.

interactions[33] (Supplementary Fig. 2c). In the structure of Zen4, Pro36 moves closer to Tyr120 (corresponding to Pro17 and His95 of kinesin-1, respectively), with the closest distance of ~6.0 Å, which would disfavour binding of any nucleotide at this site. Occlusion of the nucleotide-binding pocket was also found in the reported apo structure of kinesin-1, in which the closest distance is ~6.3 Å (ref. 22).

**Unwound α6 and backward-docked conformation of the NIS.** Unlike the previous apo structure[22], the NIS and the preceding α6 helix of Zen4 adopt new conformations that have never been seen before (Fig. 2). In the ADP-bound structure of kinesin-1, α6 forms a short helix[33]. In the ATP-bound state, α6 has one more helical turn at the C terminus due to the forward docking of the NL[17]. In the previous apo structure of kinesin-1, α6 adopts a conformation very similar to that in the ADP state[22]. In contrast, α6 of Zen4 is partially melted and ends at Phe419 (corresponding to Phe318 in kinesin-1), which is almost one helical turn shorter than the previous structure[22] (Fig. 2a; Supplementary Fig. 3a). Our structure is somewhat similar to a recent cryoEM model of kinesin-1 (PDB code 3J8X), which also showed melting of the α6 helix in the apo state[19] (Supplementary Fig. 3b).

After α6, the sequence of Zen4 makes a sharp turn at Ala420. The NIS, starting from Glu421, flips backward, forms a five-residue β-strand and becomes disordered after Ile426. The backward flipping of NIS mainly occurs through backbone hydrogen-bond interactions with the N-terminal β1c strand, becoming a part of the N-terminal antiparallel appending β-sheet (Fig. 2c,d; Supplementary Fig. 2d). This backward-docked conformation of NIS had not previously been observed.

Several lines of evidence suggest that the melting of α6 and the backward-docked conformation of the NIS is not specific to Zen4, but instead a general feature of kinesin in the apo state. First, the sequence and structure of the N-terminal appending β-sheet (β1a–c), which plays a key role in the stabilization of the backward flipping conformation of NIS, are highly conserved across the kinesin superfamily proteins (Fig. 2d; Supplementary Fig. 1). Second, an earlier study using single-molecule fluorescence resonance energy transfer (smFRET) suggested the NL of kinesin-1 was not totally disordered in the apo state, and might adopt a backward-extending configuration[34]. Third, recent cryo-EM analyses showed that the NISs of kinesin-1 and kinesin-3 in the apo state are directed to the minus end of microtubules[18]. These results suggest the NL of kinesin-1 in the apo state might adopt a conformation similar to the structure of Zen4.

**Labelling the NL of kinesin-1 for smFRET analysis.** To directly probe the conformation of the NIS of kinesin-1, we placed one fluorophore at the NIS (Thr332) and another one at the catalytic core (Glu222) of *Drosophila* kinesin-1, and performed smFRET analyses. Human kinesin-1 labelled at the corresponding positions (Thr325 and Glu215) is active[34]. To avoid problems associating with cysteine-light mutants[6], we engineered the kinesin-1 gene to encode unnatural amino residues (*p*-azidophenylalanine, pAzF) at the designed positions, which can selectively react with the DBCO-sulfo-Cy3 and DBCO-sulfo-Cy5 fluorescent dyes[35] (Supplementary Fig. 4). A hetero-dimeric kinesin-1 was made, in which only one head containing the engineered mutations was labelled, so that we could monitor the FRET signals within one head. Kinesin

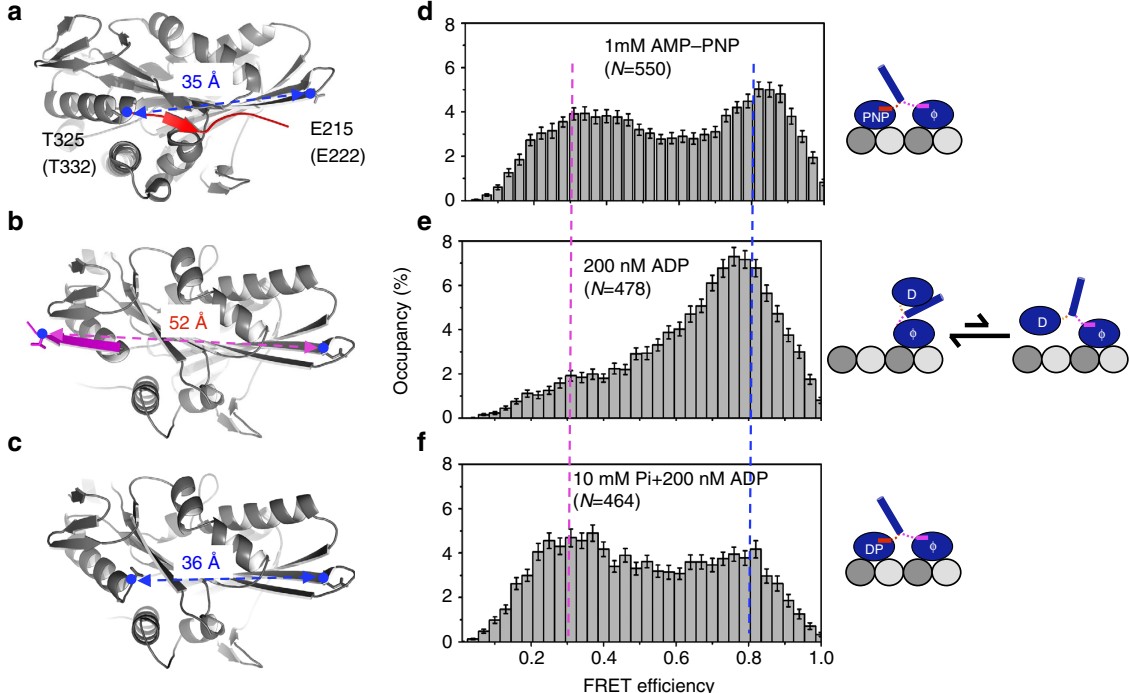

**Figure 3 | smFRET analysis of the conformational changes of NIS of kinesin-1 in different nucleotide states.** (**a**) Structure of human kinesin-1 in ATP-bound state (PDB code 3J8Y) with the forward-docked NL coloured red. The positions of the residues for dye labelling are shown in blue. The numbers in the parentheses show the equivalent residues of *Drosophila* kinesin-1. The FRET pair spans a distance of ∼3.5 nm. (**b**) Proposed structure of kinesin-1 in apo state with a backward-docked NIS (in magenta) as in the structure of Zen4. The FRET pair spans a distance of ∼5.2 nm. (**c**) Structure of kinesin-1 in ADP-bound state (PDB code 1BG2). The FRET pair spans a distance of ∼3.6 nm. (**d**–**f**) Histograms of FRET efficiency of the dye-labelled kinesin bound to microtubule in 1 mM AMP–PNP, 200 nM ADP and 10 mM Pi + 200 nM ADP, respectively. The numbers in parentheses show the molecules analysed. The dotted lines illustrate peaks characteristic of the high FRET (blue) and low FRET (magenta) signals. The cartoon diagrams show the predicted models of kinesin at the corresponding nucleotide conditions. Red and magenta bars illustrate the forward and backward-docked conformations of the NL, respectively. The double arrows in (**e**) illustrate the equilibrium between disordered and backward-docked NL. Error bars indicate s.d. of 1,000 bootstrap samples of the FRET traces.

molecules that contained both Cy3 and Cy5 were selected for smFRET analysis with total-internal-reflection fluorescence microscopy (TIRFM) as described before[36]. Consistent with the previous study[34], *Drosophila* kinesin-1 labelled at the NIS (Thr332) and the catalytic core (Glu222) remained active (Supplementary Fig. 4e,f).

**The conformations of NL of kinesin-1 in 2HB and 1HB.** In the presence of the non-hydrolysable ATP analog (AMP–PNP), kinesin is thought to bind to microtubules in the 2HB state[7,34]. Our model predicts a bimodal FRET distribution in this state, with one high and one low FRET peak. The high FRET peak corresponds to the FRET pair located in the trailing head (ATP-bound state), which spans a distance of ∼35 Å (ref. 17; Fig. 3a). The low FRET peak corresponds to the sensor pair located in the leading head (apo state), which spans a longer distance of ∼52 Å due to the melting of α6 and the backward-docked conformation of the NIS as in Zen4 (Fig. 3b). In contrast, the previous apo model for kinesin-1 shows the α6 remaining intact and the NIS adopting a disordered conformation similar to the ADP-bound state[22] (Fig. 3c). This structure predicts a unimodal distribution in the 2HB state, with a single high FRET peak corresponding to the FRET pairs in both the leading and trailing heads spanning a similar distance of ∼35 Å. Thus, the FRET efficiency in 2HB reports the NL conformations of the two heads, and provides the key information to distinguish between these two models.

The FRET signal in the presence of AMP–PNP showed a bimodal distribution, with a high FRET peak at ∼80% and a low FRET peak at ∼30%, Fig. 3d. The high and low FRET efficiencies are consistent with Thr332 in the NIS and Glu222 at the tip of motor domain spanning a distance of ∼35 and ∼52 Å in the trailing and the leading heads, respectively. Thus, the bimodal distribution in 2HB supports our model, and argues against the previous model. In particular, the low FRET peak suggests the rearward tension stabilizes a state in which the end of α6 is melted and the NL of the leading head is docked backward, consistent with the idea that the NIS of the apo head of kinesin-1 in 2HB adopts a conformation similar to the structure of Zen4.

We next probed the conformation of the NL in the absence of tension by examining the smFRET efficiency in the one-head bound state (1HB). In the presence of low concentrations of ADP, kinesin binds to microtubules with a bound head in the apo state and a tethered head in the ADP-bound state[7,37]. The FRET histograms at 200 nM ADP clearly showed decreased occupancy of the low FRET peak and increased occupancy of the high FRET peak, with a dominant peak at ∼80% (Fig. 3e). It seems that the majority of the bound apo heads have their NL disordered in the absence of intramolecular tension and adopt a conformation similar to the high-FRET ADP-bound state, as suggested before[14,22]. Interestingly, a small low FRET shoulder peak at ∼30% was observed, suggesting some portion of the bound apo heads adopt the backward-docked conformation even in the absence of tension, consistent with the backward extended conformation[7,18,34]. The data suggest that there is equilibrium

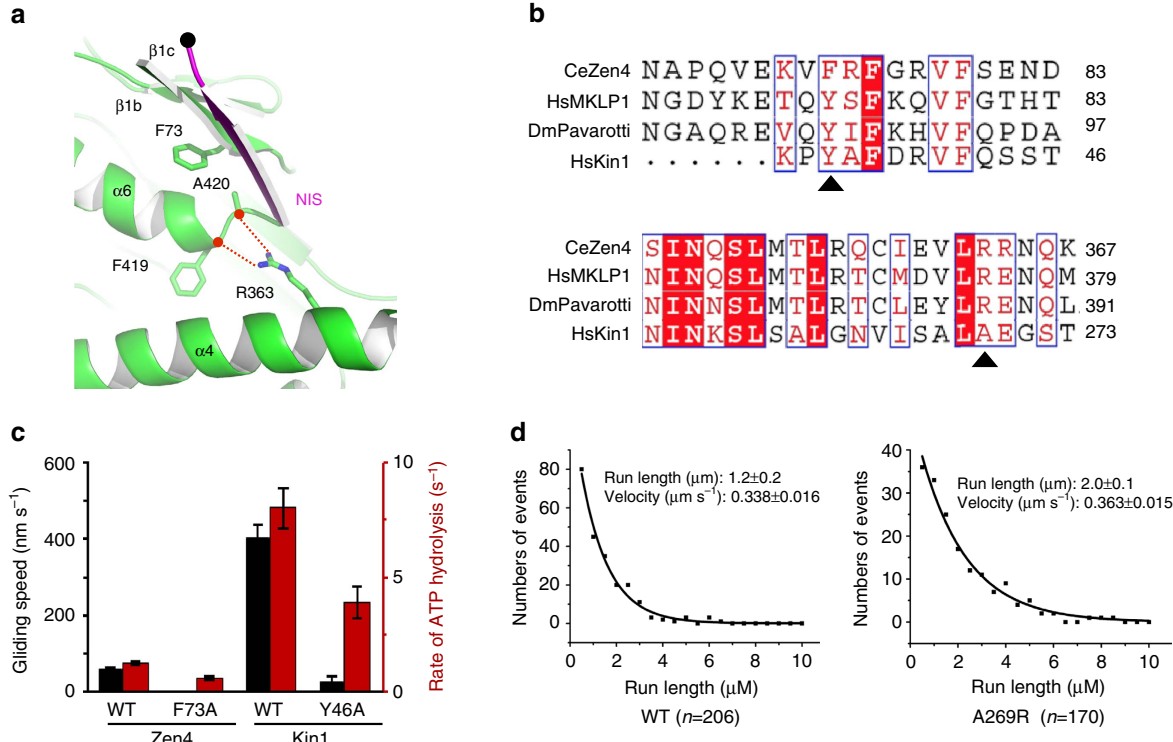

**Figure 4 | Unwound α6 and backward-docked conformation of NIS are essential for the motility of kinesin.** (**a**) Interactions that stabilize the melting of α6 and backward-docked NIS of Zen4. Hydrogen-bond interactions are showed as red dotted lines. (**b**) Sequence alignments around the β1C (top panel) and α4 (bottom panel) regions of kinesin-6 family proteins and human kinesin-1. The triangles indicate the positions of the residues mutated in this study. (**c**) ATPase and microtubule gliding activities of the WT and β1C mutants of Zen4 and rat kinesin-1. Black and red bars indicate the rate of microtubule gliding and ATP hydrolysis, respectively. Error bars indicate standard deviations for at least three independent measurements. (**d**) Run lengths of WT (left panel) and the A269R mutant (right panel) rat kinesin-1 from TIRF assays. Data were fit to a single exponential. *n* indicates the total number of molecules scored in the assays.

between disordered (high FRET) and backward-docked (low FRET) NL in the bound apo head in the 1HB state, which favours disordered as indicated by the relative height of the high/low FRET peaks.

We then added 10 mM inorganic phosphate to 200 nM ADP, which has been shown to shift the kinesin heads into the 2HB state[7]. As expected, the bimodal distribution pattern of the FRET peaks appeared (Fig. 3f). The data suggest the trailing head adopts an ADP-Pi state under these conditions, further supporting the notion that intramolecular tension in the 2HB state favours melting of α6 and a backward-docked conformation of the NIS in the leading head.

The backward-docked conformation of NIS during its ATPase cycle was also supported by cysteine-light mutants of kinesin-1 using two different FRET sensor pairs (Supplementary Fig. 4g,j). Similar to the mutant labelled with unnatural amino residues, the mutants labelled through cysteine residues remained active in ATP hydrolysis and motility.

**Backward-docked NIS is important for motility of kinesins.** To test the importance of the pairing of NIS with β1c in the apo state, we disrupted their interactions by perturbing of the structure of β1c. In the structure of Zen4, β1c makes hydrophobic contacts to α6, particularly through Phe73 (Fig. 4a). Phe73 of Zen4 corresponds to Tyr46 in mammalian kinesin-1 (Fig. 4b; Supplementary Fig. 1). We made an F73A mutation of Zen4, and then tested the motility of the mutant using microtubule-gliding assays. Whereas microtubules glided at ~56 nm s$^{-1}$ with wild

type (WT) Zen4, the F73A mutant completely lost its motility (Fig. 4c; Supplementary Fig. 5a), suggesting the N-terminal appending β-sheet plays an important role in the conformational cycle of the motor. The role of β1c in Zen4 also extends to kinesin-1—microtubules glided at ~400 nm s$^{-1}$ with WT kinesin-1, but the equivalent mutation (Y46A) reduced the velocity by a factor of 10 (Fig. 4c; Supplementary Fig. 5b). Consistent with this loss of motility being specific to β1c and not resulting a general destabilization of the motor structure, the mutations reduced ATPase activity of Zen4 and kinesin-1 only by a factor of two. Thus, the mutations of β1c decoupled ATP hydrolysis from motility, consistent with the idea that the NIS of the leading head in the 2HB state is not passively extended to the minus-end of microtubule, but actively involved through the interactions with the N-terminal appending β-sheet.

We hypothesize that the stable backward-docked conformation of the NIS was not seen in previous kinesin-1 structures due to the conformation of the C-terminal end of α6 helix[18,19,22]. In Zen4, the melting of α6 is facilitated by Arg363 from α4, which forms two hydrogen bonds with the main-chains of Phe419 and Ala420, blocking further extension of α6 (Fig 4a). We termed Arg363 the 'arginine gate' of Zen4. Arg363 is conserved in the kinesin-6 protein family but absent in many of the other kinesins (Supplementary Fig. 1). As suggested by the smFRET analysis above and proposed before[27], the melting of α6 may be facilitated by the rearward tension, which is absent in the previous structural work.

To test the importance of α6 melting, we introduced an 'arginine gate' into rat kinesin-1 (A269R). Our model predicts

that the 'arginine gate' facilitates the melting of α6, which would functions as 'super front head gating' to promote kinesin to entry the 2HB state and enhance processivity. Consistent with our prediction, introducing the A269R 'arginine gate' mutation increased the rat kinesin-1 run length from a WT value of 1.2–2.0 μm (Fig. 4d). The mutation had little effect on velocity. Because A269R lies on the surface of α4 distal to microtubule and is partially buried inside the motor domain, it is unlikely that the enhanced processivity by the A269R mutation results from nonspecific electrostatic interactions with the negatively charged microtubule. Likewise, introduction of the 'arginine gate' mutation (A276R) also enhanced the processivity of the *Drosophila* kinesin-1 (Supplementary Fig. 5c). In particular, the motility of the shortened neck-linker mutant, which has been showed to lose its processivity due to the deletion of one residue from the neck linker[38], was rescued to a level comparable to the WT motor. These data suggest that melting of α6 promotes the backward docking orientation of the NIS and the stepping cycle of the motor.

## Discussion

The ATPase cycle of kinesin consists of three principle states: ATP, ADP and apo states. It has been shown that the NL changes its conformation during the stepping cycle, between a forward-docked conformation in ATP-bound state and a disordered conformation in ADP-bound state[14]. Our structure of Zen4 shows that the NIS adopts a backward-docked conformation in the apo state. smFRET analysis suggests the NIS of kinesin-1 adopts a similar conformation under tension, instead of being disordered as generally believed before. This new conformation of the NL provides important structural insights into the stepping cycle of kinesin.

2HB is the key stepping intermediate of kinesin. We modelled the leading head in the 2HB state by replacing the previous apo structure of kinesin (PDB code 4LNU) with the structure of Zen4, whereas the trailing head was taken from the cryo-EM structure of kinesin in complex with microtubule (PDB code 3J8Y; Fig. 5a). In this model, the end-to-end distance of the NL is ~35 Å (Fig. 5b). By contrast, the NL distance based on the earlier model is ~53 Å (ref. 22). The shortening of the NL distance by ~18 Å would significantly reduce the intramolecular tension between the two bound heads, which provides the structural basis of the stability of 2HB.

The magnitude of inter-head tension has been debated. Force-clamp optical trapping experiments argued that the tension is as low as 4–6 pN, or as high as 26 pN (refs 6,10,16). To assess the magnitude of the intramolecular tension in the 2HB state, we used the WLC model for the unstructured NL regions. Based on our structural model, the calculated tensions vary between 4 and 13 pN within the window of $L_p = 0.5$–1.5 nm (Fig. 5c). At $L_p = 1$ nm, a value consistent with the measurements on multiple polypeptide chains[39,40], the intramolecular tension $F$ was estimated to be ~6.7 pN (Fig. 5c). This tension is consistent with the previous estimate[10,16], and close to the unbinding/stall force (~7 pN; refs 28,30). By contrast, the tensions based on the earlier model is much larger, varying from 13–42 pN (Fig. 5c). Specifically, the estimated tension is ~20 pN at $L_p = 1$ nm. Thus, the unwound α6 helix and the backward-docked NIS of the leading head relax the intramolecular tension to a level approaching to the stall/unbinding force, much lower than what was proposed before. This provides the rationale for kinesin to stably bind to microtubule in the 2HB state.

Consistent with this notion, the ATP/ADP-Pi-bound head is suggested to unbind from the microtubule at a rate on the order of 100 per s under the tension of ~7 pN (refs 8,41). According to the Bell's model[42], the tension of ~20 pN would probably unbind

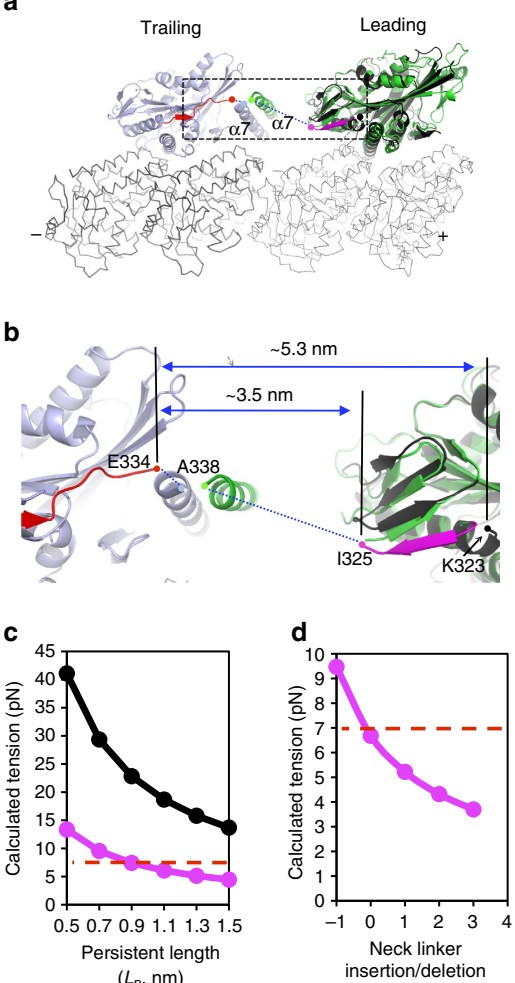

**Figure 5 | Model and the estimated intramolecular tensions in the 2HB state. (a)** Overlay of the current and previous models of kinesin in the 2HB state. These two models differ at the leading heads, which are coloured green (modelled with Zen4) and black (modelled with kinesin-1, PDB code 4LNU), respectively. The trailing head (modelled with kinesin-1, PDB code 3J8Y), tubulin, backward-docked NIS (current model) and forward-docked NL are coloured light blue, grey, magenta and red, respectively. The neck helixes are labelled α7 (starting from Ala338, PDB code 3KIN) and the disordered neck linkers are showed as dotted lines. The boxed region is zoomed up for further analysis in **(b)**. In the current model, the last ordered residues at the C-termini of the leading and trailing heads are ILe325 (magenta dot) and Glu334 (red dot), respectively, which span a distance of ~3.5 nm with a total of 17 disordered residues (defined by the end of α6 and the start of α7). In the previous model, the last ordered residues of the leading is Lys323 (black dot), and connects to the trailing head over a distance of ~5.3 nm and with a total of 19 disordered residues. **(c)** Intramolecular tensions of kinesin-1 in the 2HB state calculated based on the WLC model with different values of $L_p$. The magenta and black lines indicate the calculated forces based on the current and previous models, respectively. The red dashed line indicates the unbinding/stall force (~7 pN). **(d)** Calculated intramolecular tension of kinesin-1 in the 2HB state with different insertion/deletion in the neck linker region. 0 indicated WT kinesin, and −1 indicated deletion of one residue in the neck linker. The calculations were done at $L_p = 1$ nm.

the kinesin head instantly (at a rate ~$10^7$ per s at a characteristic distance $d = 4$ nm)[30].

Our model also provides a structural basis for understanding the dependence of the kinesin processivity on the NL length.

It has been shown that deletion of one residue in the unstructured region of the NL of kinesin-1 reduces the run length to below detection limit, whereas insertion of a few residues into the NL gradually reduces kinesin processivity[6,38].

Our model predicts that deletion of one residue increases the tension to ~9.5 pN (Fig. 5d). The increase of tension leads to rapid unbinding of the trailing head (in the ATP/ADP-Pi-bound state), which may occur even before ATP hydrolysis and phosphate release, thus disrupting the mechano-chemical cycle of kinesin. As discussed above, the tension of ~9.5 pN would increase the unbinding rate >10-fold (to ~1,000 per s), which is much faster than the rate of ATP hydrolysis and phosphate release (~100 per s; refs 6,43). Thus, our model provides a tension-based mechanism to explain the large impact of removing a single unstructured residue from the neck linker[38]. Interestingly, the gain-of-function 'arginine gate' mutation rescued the motility of the shortened NL kinesin-1, suggesting a stabilized 2HB state due to the enforced backward-docked NL in the leading apo head.

Likewise, lengthening of the NL reduces the intramolecular tension (Fig. 5d), which may slow the unbinding rate of the rear head, leading to loss of processivity (rear-head gating mechanism)[11]. Alternatively, the reduced tension may diminish processivity by weakening the front-head gating mechanism via releasing the occlusion of the nucleotide-binding pocket of the apo front head.

Our study suggests that the stretching of NL of kinesin-1 generates an intramolecular tension of ~7 pN. Interestingly, this magnitude of tension is comparable with the stall/unbinding force[28–30], which might be close to the upper limit set by the required stability of the 2HB structure and the associated chemical reactions. Thus, the optimal NL length and intramolecular tension may reflect the tight coupling of the mechanic movement to ATP hydrolysis of the motor.

ZEN4 is notable in having a long neck linker[44], and the inter-head tension in 2HB is expected to be low. The presence of the 'arginine gate' in Zen4 would further increase the stability of 2HB. Consistent with this structural divergence, Zen4 does not function as a typical transport kinesin, but instead acts as a microtubule bundling factor responsible for the formation of central spindles during cytokinesis[31]. The sequence differences that lead to more stable backward docking of the NL in Zen4 may be an evolutionary adaptation to compensate for the very long NL in this motor. By uncovering this structural adaptation, we also find that it plays a role in kinesin-1 and hence is a general mechanism, albeit one that is accentuated in ZEN4 and potentially other motors with longer NL.

## Methods

**Protein works.** The Zen4(1–601) construct (Z601) was subcloned into a pMal-p2 vector that had been modified to contain MBP tag after the start codon and a cleavage site for the TEV protease. The protein was overexpressed in the BL21(DE3) strain of E. coli as MBP-fusion proteins. It was induced with 1 mM IPTG for 12 h at 18 °C. Cells were lysed in 10 mM Hepes, 200 mM NaCl, 2 mM EDTA, 2 mM dithiothreitol (DTT), 1 mM PMSF, pH 7.0, at 4 °C using a nano homogenize machine (ATS). After centrifugation at 18,000 r.p.m. for 60 min, the supernatant was loaded onto an Amylose column, washed, and then eluted with elution buffer (200 mM NaCl, 10 mM Hepes, 2 mM DTT, 2% maltose (w/v), pH 7.0). The fusion tag was removed by treatment with TEV protease at 4 °C for ~24 h. The protein was further purified using Source 15S chromatography, and subjected to a Superdex200 column equilibrated with protein buffer (200 mM NaCl, 10 mM Hepes, 2 mM DTT, pH 7.0), concentrated to ~6 mg ml$^{-1}$ and stored at −80 °C. The mutant Z601(F73A) was purified using the same protocol. The construct of Zen4(1–441) was cloned and purified as above, except the protein was concentrated to ~20 mg ml$^{-1}$ and stored at −80 °C.

The gene of kinesin-1 KIF5B(1–560) was cloned from cDNA of Rattus norvegicus (K560), and then inserted after the NdeI restriction endonuclease site into pET-21b (His-tag) vector. The protein was overexpressed in E. coli strain BL21(DE3) and induced by 0.5 mM isopropyl b-D-thiogalactoside (IPTG) at 18 °C overnight. Cells were spun down at speed 4,000 r.p.m. (Beckman, Rotor JA4.2) for 20 min, and lysed in 10 mM Tris-HCl, 200 mM NaCl, 0.5 mM ATP, 1 mM MgCl$_2$ and 1 mM PMSF, pH 8.0, using a nano homogenize machine (ATS) at 4 °C. After centrifugation at speed 18,000 r.p.m. (Beckman, Rotor JA20) for 1 h, supernatant was separated and loaded onto a gravity column with Ni-NTA beads. The protein was washed, and eluted with elution buffer (200 mM NaCl, 10 mM Tris-HCl, 2 mM DTT, 0.5 mM ATP, 1 mM MgCl$_2$, 250 mM imidazole, pH 8.0). The protein was further purified by gel-filtration (Superdex200, GE Healthcare) chromatography. The purified protein was concentrated to 5 mg ml$^{-1}$ in 200 mM NaCl, 10 mM Tris-HCl, 2 mM DTT, 0.5 mM ATP, 1 mM MgCl$_2$, pH 8.0. The mutant of K560(Y46A), was generated by Quickchange mutagenesis. The expression and purification method was the same as above.

The encoding sequence of DmKhc(1–401) was cloned from cDNA of Drosophila melanogaster, and inserted after the BamH1 restriction site in a modified pCDF-duet (His-tag) vector that had been modified to contain a cleavage site for the TEV protease before the gene. The WT protein was overexpressed and purified as above. The purified protein was concentrated to ~15 μM in 150 mM NaCl, 10 mM Hepes, 0.25 mM ATP, 1 mM MgCl$_2$, 2 mM DTT, pH 7.0 and stored at −80 °C.

For smFRET experiment, the encoding sequence of DmKhc(1–401) was inserted after the NdeI restriction site in a modified pMAL-p2 (MBP-tag) vector that had been modified to contain a cleavage site for the TEV protease before the gene. To insert an unnatural amino acid p-azidophenylalanine (pAzF), we constructed the mutant DmKhc (1–401) by substituting Glu222 and Thr332 with the amber codon (TAG)[35]. The C-terminal of the mutant protein was followed by a spacer (PGGS) and a strep tag.

To express the proteins with pAzF, the pMAL-p2 vector carrying the mutant kinesin head and the pCDF-duet vector carrying the WT kinesin head were co-transformed with pEVOL-pAzF (Addgene) into E. coli BL21(DE3). When the cells reached OD600 ~0.6, 1 mM IPTG and 0.02% L-Arabinose (final concentration) were added to induce the protein expression, and pAzF was also added to a final concentration of 1 mM. The protein was expressed at 18 °C overnight. Cells were lysed in 10 mM Tris-HCl, 200 mM NaCl, 1 mM EDTA, 1 mM dithiothreitol (DTT), 1 mM PMSF, pH 8.0, at 4 °C using a nano homogenize machine (ATS). After centrifugation at 18,000 r.p.m. for 60 min, the supernatant was loaded onto a gravity column with Ni-NTA beads, washed and eluted with elution buffer (200 mM NaCl, 10 mM Tris-HCl, 2 mM DTT, 250 mM imidazole, pH 8.0). The protein was then loaded onto a Strep-Tactin column, washed and eluted with elution buffer (150 mM NaCl, 10 mM Tris-HCl, 0.05 mM ATP, 0.05 mM MgCl$_2$, 2.5 mM desthiobiotin, pH 8.0). The fusion tag was removed by treatment with TEV protease at 4 °C for 12–24 h. The protein was further purified using Source 15Q chromatography, and subjected to a Superdex200 column equilibrated with protein buffer (150 mM NaCl, 10 mM Hepes, 0.25 mM ATP, 1 mM MgCl$_2$, 2 mM DTT, pH 7.0), concentrated to ~5 μM and stored at −80 °C.

To generate the cysteine-light (CL) mutant, mutations (C17V, C45S, C338S) were introduced into the Drosophila kinesin-1 (1–401) by Quickchange. Purification of the CL mutant was the same as the WT protein. To achieve selective labelling, we constructed a 227/332-CL mutant by substituting Q227 and T332 with cysteine in the context of the CL mutant. To express the heterodimeric mutant proteins, a pMAL-p2 vector carrying 227/332-CL and pCDF-duet vector carrying the CL mutant were co-transformed into E. coli BL21(DE3). The protein was purified similarly as described above. Through a similar protocol, we obtained the 128/332-CL mutant.

**Crystallization and structure solution.** Crystals of the motor domain of Zen4 (1–441) were grown at 18 °C by hanging-drop vapour diffusion methods. The native crystals grew from 1.45–1.55 M (NH$_4$)$_2$SO$_4$, 100 mM MES, 200 mM NaI, 4% dioxane, 10 mM DTT, pH 6.5. All crystals were harvested from buffer (75 mM NaCl, 1.6 M (NH$_4$)$_2$SO$_4$, 50 mM MES, 150 mM NaI, 4% dioxane, pH 6.5) and flash-frozen in liquid nitrogen. Diffraction data were collected at −170 °C at the beamline of Shanghai Synchrotron Radiation Facility (SSRF), and they were processed with the HKL2000.

The structure was solved using molecular replacement with Phaser. The resolution cutoff was selected based on CC1/2 = 0.5 (ref. 45). The structure of kinesin-1 (PDB code 1BG2) was used as the search mode[33]. After the initial search, the model was completed manually using Coot. The final model was refined with Phenix, with $R_{work} = 0.20/R_{free} = 0.23$, Molprobity score 1.49 (100%) and Clash score 7.04 (99%) (ref. 46).

**Microtubules-activated ATPase assays.** All reagents for the MESG-based microtubules-activated ATPase assay were obtained from Cytoskeleton. Reactions were set up in wells of a 96-well plate (Corning Costar No. 3697) and each well contained 50 mM Tris-HCl, pH 7.5, 1 mM MgCl$_2$, 0.1 mM sodium azide, 20 μM paclitaxel, 0.5 μM MT, 0.5 mM ATP, 0.1 unit purine nucleoside phosphorylase2 (PNP), 0.2 mM MESG reagent, and kinesin proteins in a reaction volume of 200 μl. NaCl (150 mM) was added to the assay conditions for WT and F73A mutant Z601. Reactions were started by the addition of ATP and were read every 10 s at 360 nm for a total of 20 min using a monochromatic spectrophotometer (SpectroMax250, Molecular Devices, San Diego, CA). The assay is based on an absorbance shift

(330–360 nm) that occurs when MESG is catalytically converted to 2-amino-6-mercapto-7-methyl purine in the presence of inorganic phosphate and PNP.

**Microtubules gliding assays.** The assays were done as described before[47]. Coverslips (12-545-F, Thermo Fisher, USA) and slides (10127101P, Shitai, Jiangsu, China) were used to make the flow chamber (containing four individual channels) for the gliding assays. Two-millimetre-wide double-sided tapes (200 MP, 3M (Minnesota Mining and Manufacturing), USA) were used to stick and isolate the flow channels. The surface was cleaned in acetone (Guoyao, China) followed by KOH (484016, Sigma-Aldrich, USA) with a sonicator (KQ500DE, Kunshan, China) and stored in water to keep it hydrophilic. A volume of 15 μl of the motor proteins (KIF5B or Zen4) were added to the flow chamber channels for 5 min. The motor-coated coverslips were blocked with 3 mg ml$^{-1}$ casein, followed by addition of 40 nM microtubules for 5 min. An energy system (0.5 mM ATP, 20 μM taxol, 10 mM DTT, and 1 mg ml$^{-1}$ casein), an ATP regeneration system, and an oxygen scavenger system were flowed into the chamber. Images of gliding assay were recorded every 500 ms using a Nikon Ti-E TIRF microscope under 640-nm laser excitation.

**Single-molecule motility assays.** For the motility assays using *Drosophila* kinesin-1 truncated at position 401, the assays were done similarly as above. Microtubule filaments were immobilized on the glass surface through antibody towards beta-tubulin. The non-specific binding was blocked by incubation with 3 mg ml$^{-1}$ casein solution for 5 min. Approximately 100 nM EGFP-fused WT rat Kinesin-1 motor domain or 50 nM the A269R mutant motor protein was flowed into the chamber, with an energy system, an ATP regeneration system, and an oxygen scavenger system. Images of motility assay were recorded every 500 ms using a Nikon Ti-E TIRF microscope under 488-nm laser excitation with a TIRF illumination mode. The images were analysed by a home-written Matlab code to grab the single molecule localization and to filter the tracks. Single-molecule motility assays with the dye-labelled DmKhc(401) were performed similarly under 647-nm laser excitation.

For the processivity rescue experiment, *Drosophila* KHC truncated at position 560 was used, with a C-terminal eGFP and His-tag. Insertions and deletions were made using Q5 (New England Biosciences). Flow cells were prepared and microtubules were deposited using rigor kinesin as reported previously[48]. Motors were diluted to 100 pM final concentration in imaging solution: 0.5 mg ml$^{-1}$ casein, 10 μM taxol, 20 mM glucose, 20 μg ml$^{-1}$ glucose oxidase, 8 μg ml$^{-1}$ catalase, 0.2 mg ml$^{-1}$ BSA, 1:200 β-mercaptoethanol, and 2 mM MgATP in BRB80 (80 mM PIPES, 1 mM EGTA, 1 mM MgCl$_2$, pH 6.8). Imaging was done under total internal reflection fluorescence microscopy using a Nikon TE2000 inverted microscope and a 488 nm Argon laser (Spectra Physics). A Cascade 512 EMCCD camera (Roper Scientific) and MetaVue software (Molecular Devices) were used to capture images at 3 frames per second. The nanometric positions of individual GFP motors was determined by point spread function fitting using FIESTA software[49]. Single-track velocity was determined by linear fitting to the distance over time trace, and single-track run length was determined by the absolute distance travelled in a trace. Population velocity was reported as the sample mean ± s.e. with a 10% error added for lack of temperature control within 1 °C. Population run length was determined by fitting the empirical cumulative density function to the exponential distribution with an X offset (runs shorter than five pixels were dropped due to underfilled bins, 71 nm per pixel). Error on run lengths was determined by bootstrapping[50]. Experiments were run at 22–23 °C.

**Labelling and smFRET measurements.** The DmKhc(1–401) mutant was labelled by incubation with DBCO-sulfo-Cy3 and DBCO-sulfo-Cy5 (Jena Bioscience) at a molar ratio of 1:5:5 (protein: Cy3: Cy5) for 1 h at room temperature in the reaction buffer (buffer A) containing 10 mM HEPES(pH7.0), 150 mM NaCl and 1 mM MgCl$_2$. The excess free dyes were removed using Zeba spin desalting columns (Thermo Scientific). We estimated the extent of the labelling from absorption spectra of labelled protein by measuring peak maxima at 532 and 650 nm for Cy3 and Cy5 using Implen nanodrop P-300. Protein concentration was determined using BCA protein assay kit (Pierce). Under the same condition, control labelling experiment with WT DmKhc(1–401) resulted <10% non-specific incorporation of Cy3 or Cy5 dyes.

The 227/332-CL and 128/332-CL mutants were labelled by incubation with Cy3-maleimide and Cy5-maleimide (GE Healthcare) for 30 min at 4 °C. The excess free dyes were removed using Zeba spin desalting columns (Thermo Scientific).

smFRET measurements were performed with an objective-based total internal reflection fluorescent (TIRF) microscope as described before[36]. In our system, objective-based TIRFM was used. In order to remove background noise, the filter sets used here are ET585/65 (Chroma) for Cy3 and single-band bandpass filter FF02-675/67-25 (Semrock) for Cy5. In Prism-based TIRF, long-pass filter ET655LP is used to collect more emission light from Cy5, which will generate higher FRET values than those measured by the objective-based TIRFM used in this study.

To calibrate the microscope, we synthesized a 14 bp DNA ladder. The N terminus of the acceptor strand (5′-(NH$_2$ C6) CATGACCATGACCAG (Biotin)-3′) contained a C6-NH$_2$ moiety and C terminus was biotinylated to allow the DNA to be labelled by Sulfo-Cy5-NHS and be immobilized on a streptavidin surface. The donor strand (5′-CTGGTCATGGTCATG-3′) contained a single amine-modified dT at 2 dT residue site and was labelled by Sulfo-Cy3-NHS. After the labelling, the

donor strand and the biotinylated strand were hybridized in the presence of 200 mM KCl by heating the solution to 75 °C followed by passive cooling to room temperature. After calculation, the distance for the 14 bp DNA ladder is about 58 Å, which provides a FRET peak ~0.35 measured with our microscope system.

The labelled kinesins were imaged in the presence of 1 mM AMPPNP, 200 nM ADP (5 U ml$^{-1}$ hexokinase converted contaminating ATP) and 200 nM ADP with 10 mM Pi, respectively. To immobilize microtubules onto glass surface, microtubules was incubated with biotin-maleimide at 10:1 (M/M) ratio for 1 h at 37 °C and then added to the streptavidin coated surface for 1 min at 1 μM and unbound microtubules were removed by washing the channel with buffer A. Dye labelled DmKhc (~10 nM) were attached onto the microtubules for around 1 min to a density where single fluorescent molecules could be clearly distinguished. Without microtubules, there is no significant surface immobilization of the kinesin. Cy3 fluorophore was excited with 532 nm laser (Coherent Inc., Sapphire SF). Photon emitted from Cy3 and Cy5 were collected using 1.49 NA ×100 objective (Olympus UAPON ×100 OTIRF), and Optosplit II (Cairn Research Limited) was used to separate spatially Cy3 and Cy5 frequencies onto a cooled EMCCD (Andor iXon Ultra). Fluorescence data were acquired using the software Metamorph (Universal Imaging Corporation). Images were taken at 50 ms per frame.

**smFRET data analysis.** The data were analysed using custom software written in MatLab (MathWorks). Cy3 and Cy5 channel were mapped using TetraSpeck fluorescent microsphere beads (Invitrogen, 0.1 μm). At least more than 10 beads were selected to get the transformation matrix used in mapping in MatLab. Photobleaching events in each trace were detected as a significant drop (≥3 times s.d. of background noise) in the median filtered (window size = 9 frames) total fluorescence intensity ($I_{total} = I_{Cy3} + I_{Cy5}$) without returning to the previous average level. Signal-to-background noise ratios are calculated as total intensity relative to the s.d. of background noise: $I_{total}$/[s.d. ($I_{Cy3}$) + s.d. ($I_{Cy5}$)]. Traces were selected automatically to meet the following criteria: a single catastrophic photobleaching event, at least 8:1 signal-to-background noise ratio, a donor-to-acceptor Pearson's correlation coefficient <0. Spectral bleed-through of Cy3 intensity on the acceptor channel was corrected by subtracting 7.5% of donor signal from the acceptor. FRET traces were calculated as: FRET = $I_{Cy5}/(I_{Cy3} + I_{Cy5})$, where $I_{Cy3}$ and $I_{Cy5}$ are the instantaneous Cy3 and Cy5 fluorescence intensities, respectively. The bin size of all histograms was set as 0.03. The data in the first second from each trace were extracted and histogram at each time point was obtained and normalized to total counts (to avoid dominant effect long traces). Traces shorter than 1 s were discarded. Contribution of the photophysical zero-FRET state in FRET histograms was removed by fitting the data to a two-state model ($E_1 = 0.1 \pm 0.1$ and $E_2 = 0.4 \pm 0.1$) with the segmental $k$-means algorithm. Error bars in FRET histograms present the s.d. of 100 bootstrap samples of each set of FRET traces examined. Every experiment was repeated in different days twice and found no significant difference.

**Intramolecular tension calculated with the WLC model.** The force ($F$) required to extend a polymer calculated with the WLC model is given by:

$$F = \frac{k_B T}{L_p} \left[ \frac{1}{4} \left(1 - \frac{x}{L_c}\right)^{-2} + \frac{x}{L_c} - \frac{1}{4} \right]$$

$k_B$ is Boltzmann's constant; $T$ is the absolute temperate; $L_p$ is the persistent length; $L_c$ is the contour length and $x$ is the end-to-end distance. The $L_c$ of the neck linker is equal to the total number of the disordered residues between the two bound heads multiplied by the distance per amino acid (0.364 nm) as described[27]. The range of $L_p = 0.5$–1.5 nm was used for the calculation, as most of the motility assays for kinesins were carried out under mild ionic strength (80 mM PIPES).

**Data availability.** Coordinates and structure factors have been deposited in the Protein Data Bank under accession numbers 5X3E. All other data are available from the corresponding author upon reasonable request.

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

## Acknowledgements

We thank Shilong Fan at the Center of Structure Biology (Tsinghua University) and the staff at beamline BL17U of SSRF for help with diffraction data collection, Haipeng Gong for discussion, Hongwei Wang and Tianyang Liu for providing microtubules, and the Tsinghua University Branch of China National Center for Protein Sciences (Beijing) for providing the facility support. This work was supported by the Chinese Key Research Plan-Protein Sciences (2014CB910100), the National Natural Science Foundation of China (31630046,31270762) and the 'Junior One Thousand Talents' program to Z.C., and grants from the National Science Foundation of China 31271423 and 863 Program SS2015AA020406 for Y.S., and National Institutes of Health Grant R01 GM076476 to W.O.H.

## Author contributions

R.G. crystallized Zen4 and performed the biochemical analyses. L.Z. and Y.Z. performed FRET analysis; Q.S. and Y.S. performed the motility analyses of Zen4, *Drosophila* K401 and rat K560; K.J.M., G.Y.C. and W.O.H. performed the motility analyses of *Drosophila* K560; Z.C. wrote the manuscript with help from all authors; Z.C. directed and supervised all of the research.

## Additional information

**Competing interests:** The authors declare no competing financial interests.

