## [Peer Review File · Nature Communications]

Reviewers' Comments:

Reviewer #1 (Remarks to the Author)

This manuscript describes the structure and analysis of the kinesin-6 motor domain of Zen4 in the nucleotide free state. The most interesting finding is that the neck linker domain is docked in backwards conformation, which has not previously been visualized. This fills in the details of a previously missing step in the kinesin mechanochemical cycle, and as such represents a very significant addition to the field. Based on the structure, the authors perform an elegant series of FRET experiments to add compelling evidence to their model, which involves a switch between the neck linker docking backwards and forwards. Prior studies suggested a switch between linker forward and linker disordered, but the addition of the linker-docked-backward state clarifies a number of findings in the field that could not previously be explained, including EM data suggesting the linker was docked backwards (in combination with a melting of helix 6) and calculations based on the tension between the two heads. The current work pulls all of this data together, creating a two-head-bound tension model that is based on the structure and FRET evidence for this new state. Finally, the authors are able to introduce an "arginine-gate" into kinesin-1, causing it to behave more like Zen4, and also mutate residues in Zen4 that disrupt the backwards-docked confirmation. Both of these experiments lend evidence that the authors have captured a biologically relevant missing state of the kinesin motor that is likely to exist in many kinesin subfamilies.

The paper is clearly written and easy to follow, a few minor suggestions/comments:

- the 2fof- ρ electron density shown for the sulfate seems too large. Is there any other candidate for this position? Have the authors looked at an fo- ρ difference map with the sulfate removed?
- It would be helpful to label Switch I in Fig 1C.
- In 1D and the discussion, what is the minimal distance between Pro36 and Tyr120 in the kinesin-1 nucleotide structure?

Reviewer #2 (Remarks to the Author)

In this paper, the authors determined a crystal structure of Zen4, which is a kinesin-6 family motor protein, and performed single molecule FRET (smFRET) experiments on *Drosophila* kinesin-1. From these two sets of experiments, they claimed to find a new kinesin conformation, in which the NIS (neck initial segment) is docked backward, and proposed that this conformation is important for the processive movement of kinesin-1.

Although the claimed new conformation is potentially important for understanding the mechanism behind the processive movement of kinesin-1, the data is insufficient to support the claim. First, the extrapolating from the Zen4 (kinesin-6) crystal structure to a different kinesin

family (kinesin-1) is not logical. Second, the smFRET experiments based on just one pair of FRET probes is not enough to support the claimed conformation and more validation is required. At this point, the reviewer does not support this manuscript to be published in Nature Communications. Detailed points are described below.

Major points:

1. Extrapolating from the Zen4 crystal structure to kinesin-1

In page 6, the authors tried to claim Zen4 crystal structures are also applicable kinesin-1, however the evidence is not conclusive. First, Zen4 and kinesin-1 have significant sequence difference in some of the key elements: the NIS sequence of kinesin-6 (Fig. 2A: ESQ) is different from kinesin-1's sequence (RAK). Second, the recent cryo-EM analysis (ref. 18) and smFRET experiment (ref. 34) just indicate the neck linker in the no-nucleotide state is "biased" backward and didn't show that the backward "docked" conformation.

2. Single molecule FRET experiments.

First, the authors showed only static FRET data, either in the AMPPNP state or in the ADP state. Since they didn't carry out microtubule affinity purification after incubating with fluorescent dyes, it is highly likely that the kinesin protein used for smFRET included inactive fractions. Therefore, they should demonstrate that the kinesins are active and can move along the microtubules.

Second, the authors used shorter kinesin-1 (410 a. a.) than used in previous studies (Tomishige et al., 2006: 490 and 560 a. a. long). It is known that the probability of dimer dissociation is higher in such shorter kinesin-1, which makes the interpretation of smFRET difficult as discussed below.

Third, the authors measured FRET efficiencies 80% and 30% and concluded that these correspond to 35 Å and 52 Å, respectively. The Foster distance (R_0) of 50% FRET between Cy3 and Cy5 is ~5.5 nm; ~5.3 nm (Tomishige et al., 2006 NSMB p893) ~5.8 nm (Zhao et al., 2010 Nature). If we assume the melting of $\square 6$ and backward-docking like the one shown in Fig. 3B, the distance between probes is ~5.2 nm, so the FRET efficiency become near 50%. But they show ~30% low FRET peak, indicating as much as 80 Å distance between the two probes. One possible explanation is that the low FRET efficiency is that the two probes are located at different heads, which are separated by ~8 nm. According to Mori et. Al. 2007, the Cy3-Cy5 FRET efficiency between two heads in AMPPNP state showed ~30% FRET peak (Mori et al 2007). Although the authors used heterodimer, of which only one head has the probe binding site, as stated in the second point, the shorter coiled-coil of kinesin-1 may cause dissociation and re-association of the kinesin polypeptide.

In any case, because FRET measurements is inaccurate in measuring the distance between the two probes, just one FRET pair is not enough to claim the new conformation. Demonstration of active kinesin protein as well as more than two pairs of FRET measurements is required.

Reviewers' comments:

Reviewer #1 (Remarks to the Author):

This manuscript describes the structure and analysis of the kinesin-6 motor domain of Zen4 in the nucleotide free state. The most interesting finding is that the neck linker domain is docked in backwards conformation, which has not previously been visualized. This fills in the details of a previously missing step in the kinesin mechanochemical cycle, and as such represents a very significant addition to the field. Based on the structure, the authors perform an elegant series of FRET experiments to add compelling evidence to their model, which involves a switch between the neck linker docking backwards and forwards. Prior studies suggested a switch between linker forward and linker disordered, but the addition of the linker-docked-backward state clarifies a number of findings in the field that could not previously be explained, including EM data suggesting the linker was docked backwards (in combination with a melting of helix 6) and calculations based on the tension between the two heads. The current work pulls all of this data together, creating a two-head-bound tension model that is based on the structure and FRET evidence for this new state. Finally, the authors are able to introduce an "arginine-gate" into kinesin-1, causing it to behave more like Zen4, and also mutate residues in Zen4 that disrupt the backwards-docked confirmation. Both of these experiments lend evidence that the authors have captured a biologically relevant missing state of the kinesin motor that is likely to exist in many kinesin subfamilies.

The paper is clearly written and easy to follow, a few minor suggestions/comments:

- the 2fof- f_c electron density shown for the sulfate seems too large. Is there any other candidate for this position? Have the authors looked at an fo- f_c difference map with the sulfate removed?

In addition to a high concentration of sulfate ion, the crystallization solution contains another type of negative charged ion, iodide. Considering its larger radius, iodine may be a better candidate for this position. In the revised manuscript, we replaced the sulfate group bound by the P-loop of Zen4 with an iodide ion, and stated that the bound ion is most likely to be iodide. We generated the Fo-Fc map with the bound ion removed (Supplementary Figure 2A), which shows ball-shaped electron density bound by the P-loop. Since there is some uncertainty about this ion, we moved this figure panel to the supplementary material. We would like to emphasize that the change of a sulfate group to an iodide ion bound by Zen4 does not alter the biological interpretation of the data, as the conclusion that the protein is in a nucleotide free, apo state remains unchanged.

- It would be helpful to label Switch I in Fig 1C.

As suggested, we labeled Switch I, and Switch II as well, in the revised manuscript. We moved this detailed analysis of the structure to the supplementary material

(Supplementary Figure 2B).

- In 1D and the discussion, what is the minimal distance between Pro36 and Tyr120 in the kinesin-1 nucleotide structure?

Based on the van der Waals radius of carbon atom (1.7 Å), the minimal distance between Pro36 and Tyr120 in the nucleotide-bound state should be ~6.8 Å, which is consistent with the corresponding distance of 7.0 Å in the structure of kinesin-1 in ADP state (PDB code 1BG2). To make this clearer, we revised the related sentence at page 5 and stated that in the ADP-bound state of kinesin-1, the adenine base of ADP is tightly sandwiched between Pro17 of L1 and His93 of α 2a through van der Waals interactions.

Reviewer #2 (Remarks to the Author):

In this paper, the authors determined a crystal structure of Zen4, which is a kinesin-6 family motor protein, and performed single molecule FRET (smFRET) experiments on Drosophila kinesin-1. From these two sets of experiments, they claimed to find a new kinesin conformation, in which the NIS (neck initial segment) is docked backward, and proposed that this conformation is important for the processive movement of kinesin-1.

Although the claimed new conformation is potentially important for understanding the mechanism behind the processive movement of kinesin-1, the data is insufficient to support the claim. First, the extrapolating from the Zen4 (kinesin-6) crystal structure to a different kinesin family (kinesin-1) is not logical. Second, the smFRET experiments based on just one pair of FRET probes is not enough to support the claimed conformation and more validation is required.

At this point, the reviewer does not support this manuscript to be published in Nature Communications. Detailed points are described below.

Major points:

1. Extrapolating from the Zen4 crystal structure to kinesin-1

In page 6, the authors tried to claim Zen4 crystal structures are also applicable kinesin-1, however the evidence is not conclusive. First, Zen4 and kinesin-1 have significant sequence difference in some of the key elements: the NIS sequence of kinesin-6 (Fig. 2A: ESQ) is different from kinesin-1's sequence (RAK). Second, the recent cryo-EM analysis (ref. 18) and smFRET experiment (ref. 34) just indicate the neck linker in the no-nucleotide state is "biased" backward and didn't show that the backward "docked" conformation.

We agree with the reviewer that the previously reported evidences for the backward docked neck-linker are not conclusive. Yet, we did not base our model solely on these earlier results, but validated our discovery through mutagenesis and smFRET analyses. We further validate our conclusion with additional evidences in the revised manuscript.

The pairing of NIS with the N-terminal appending β -sheet is achieved through main-chain H-bond interactions, with no specific side-chain involved. Thus, the difference in the NIS sequences would probably not perturb the pairing interaction too much. At the second paragraph at page 6, we talk about the conservation of the N-terminal appending β -sheet (β 1a-1c), but not NIS. We clarify this point in the revised manuscript.

We did not claim that the previous smFRET and cryoEM analyses showed a backward docked conformation. They showed "backward-extending" conformation. It is our work that uncovered the backward docked NIS. As discussed above, the previous observations are supportive, but not conclusive.

2. Single molecule FRET experiments.

First, the authors showed only static FRET data, either in the AMPPNP state or in the ADP state. Since they didn't carry out microtubule affinity purification after

incubating with fluorescent dyes, it is highly likely that the kinesin protein used for smFRET included inactive fractions. Therefore, they should demonstrate that the kinesins are active and can move along the microtubules.

In the submitted manuscript, we have explicitly stated that the labeled kinesin was active at the last sentence of the first paragraph at page 7 (data shown in Supplementary figure 4E and 4F in the revised manuscript), with run length of $\sim 0.54 \mu\text{m}$ and run speed of $0.17 \mu\text{m/s}$. Likewise, we demonstrate that the kinesin mutants with the new label schemes are also active (Supplementary figure 4G and 4I) in the revised manuscript.

Second, the authors used shorter kinesin-1 (410 a. a.) than used in previous studies (Tomishige et al., 2006: 490 and 560 a. a. long). It is known that the probability of dimer dissociation is higher in such shorter kinesin-1, which makes the interpretation of smFRET difficult as discussed below.

Shorter constructs of kinesin-1 have been widely used to study the motility of the motor (Andreasson et al. eLIFE 2015; Martin et al. PNAS 2010; Fehr et al. Biophysical Journal 2009; Hackney and Stock Biochemistry 2008; Rosenfeld et al. JBC 2002). A shorter construct of kinesin-1 (401 aa) has been shown to have comparable run velocity as the longer construct (559 aa), particularly under no external load (Andreasson et al. eLIFE 2015).

Third, the authors measured FRET efficiencies 80% and 30% and concluded that these correspond to 35 Å and 52 Å, respectively. The Foster distance (R_0) of 50% FRET between Cy3 and Cy5 is $\sim 5.5 \text{ nm}$; $\sim 5.3 \text{ nm}$ (Tomishige et al., 2006 NSMB p893) $\sim 5.8 \text{ nm}$ (Zhao et al., 2010 Nature). If we assume the melting of $\square 6$ and backward-docking like the one shown in Fig. 3B, the distance between probes is $\sim 5.2 \text{ nm}$, so the FRET efficiency become near 50%. But they show $\sim 30\%$ low FRET peak, indicating as much as 80 Å distance between the two probes. One possible explanation is that the low FRET efficiency is that the two probes are located at different heads, which are separated by $\sim 8 \text{ nm}$. According to Mori et. Al. 2007, the Cy3-Cy5 FRET efficiency between two heads in AMPPNP state showed $\sim 30\%$ FRET peak (Mori et al 2007). Although the authors used heterodimer, of which only one head has the probe binding site, as stated in the second point, the shorter coiled-coil of kinesin-1 may cause dissociation and re-association of the kinesin polypeptide.

FRET efficiency is related to several issues and depends on the microscope filter set up. In our system, we use an objective based TIRFM. In order to remove background noise, the filter sets used here are ET585/65 (Chroma) for cy3 and single-band bandpass filter FF02-675/67-25 (Semrock) for Cy5. In a prism based TIRF, long pass filter ET655LP is used to collect more emission light from Cy5, which will generate higher FRET values than those measured by the objective based TIRFM used in this study.

To calibrate the microscope, we synthesized a 14 bp DNA ladder. The N-terminus of the acceptor strand (5'-(NH₂ C6) CATGACCATGACCAG (Biotin)-3') contained a C6-NH₂ moiety and C-terminus was biotinylated to allow the DNA to be labeled by Sulfo-cy5-NHS and be immobilized on a streptavidin surface. The donor strand (5'-CTGGTCATGGTCATG-3') contained a single amine-modified dT at 2 dT residue site and can be labeled by Sulfo-cy3-NHS. After the labeling, the donor strand and the biotinylated strand were hybridized in the presence of 200 mM KCl by heating the solution to 75 °C followed by passive cooling to room temperature. After calculation, the distance for the 14 bp DNA ladder is about 58 angstrom. The FRET peak measured with our microscope is ~0.35. We include this method section in the revised manuscript.

Since the absolute FRET value is sensitive to several issues, and also dependent on the labeling sites. We assign the low FRET value of 0.3 to the distance of 5.2 nm in Fig. 3D.

In any case, because FRET measurements is inaccurate in measuring the distance between the two probes, just one FRET pair is not enough to claim the new conformation. Demonstration of active kinesin protein as well as more than two pairs of FRET measurements is required.

In the revised manuscript, we provide additional validation of our arguments with two new FRET sensor pairs (supplementary figure 4G-4J). Similar as before, the new FRET data show two peaks under the AMP-PNP condition, and the FRET distributions shift to one high FRET peak in the presence of ADP.

Furthermore, we validate the observed melting of $\alpha 6$ and backward docked NIS by introduction of the “arginine gate” (A276R) into a long construct of drosophila kinesin-1 mutant (1-560 aa). As observed before, the “arginine gate” mutation enhances the processivity, and even rescues the motility of the neck-linker shortened mutant, which has been showed to lose its processivity due to the deletion of one residue from the disordered neck linker region (Shastry et al. Curr. Biol. 2010). These new data are shown in the supplementary figure 5C in the revised manuscript.

Reviewers' Comment:

Reviewer #1 (Remarks to the Author):

The authors have addressed the minor concerns I previously expressed. I remain very enthusiastic about this work.

Reviewer #2 (Remarks to the Author):

Satisfied with the revisions